# Sources and upstream pathways of the densest overflow water in the Nordic Seas

Jie Huang [1,2 ✉], Robert S. Pickart[2], Rui Xin Huang [2], Peigen Lin [2], Ailin Brakstad [3] & Fanghua Xu[1,4]

Overflow water from the Nordic Seas comprises the deepest limb of the Atlantic Meridional Overturning Circulation, yet questions remain as to where it is ventilated and how it reaches the Greenland-Scotland Ridge. Here we use historical hydrographic data from 2005-2015, together with satellite altimeter data, to elucidate the source regions of the Denmark Strait and Faroe Bank Channel overflows and the pathways feeding these respective sills. A recently-developed metric is used to calculate how similar two water parcels are, based on potential density and potential spicity. This reveals that the interior of the Greenland Sea gyre is the primary wintertime source of the densest portion of both overflows. After subducting, the water progresses southward along several ridge systems towards the Greenland-Scotland Ridge. Kinematic evidence supports the inferred pathways. Extending the calculation back to the 1980s reveals that the ventilation occurred previously along the periphery of the Greenland Sea gyre.

[1] Ministry of Education Key Laboratory for Earth System Modeling, and Department of Earth System Science, Tsinghua University, Beijing, China. [2] Woods Hole Oceanographic Institution, Woods Hole 02543, USA. [3] Geophysical Institute, University of Bergen and Bjerknes Centre for Climate Research, 5007 Bergen, Norway. [4] Southern Marine Science and Engineering Guangdong Laboratory, Zhuhai, China. ✉email: huangj15@mails.tsinghua.edu.cn

The overflows from the Nordic Seas feed the lower limb of the Atlantic Meridional Overturning Circulation (AMOC), which helps regulate Earth's climate. There are two general classes of overflow water:[1] (1) "Atlantic-origin" overflow water which is formed by strong air-sea heat loss along the rim current system of the Nordic Seas;[2] and (2) "Arctic-origin" overflow water which stems from the interior basins of the western Nordic Seas where water mass transformation takes place via open-ocean convection[3]. The East Greenland Current (EGC) transports Atlantic-origin water to Denmark Strait, while the North Icelandic Jet (NIJ) advects a comparable amount of the denser Arctic-origin water to the strait[3–5] (Fig. 1). The leading hypothesis is that the NIJ sources from the Iceland Sea as the lower limb of a local overturning cell[3]. Both types of overflow water also enter the North Atlantic through the Faroe Bank Channel[6,7]. Previous studies suggest that the deep water formed in the Greenland Sea, and a local overturning loop within the Norwegian Sea, supply this overflow[8,9]. To date, however, consensus has not been reached regarding the origin and upstream pathways of the densest component of the overflow water, which ventilates the deepest layers of the North Atlantic.

Water denser than 27.8 kg m$^{-3}$ (potential density referenced to the sea surface, $\sigma_0$) is generally identified as overflow water[10]. The Arctic-origin overflow water, found in the deepest part of Denmark Strait and the Faroe Bank Channel, is characterized by $\sigma_0 \geq$ 28.03 kg m$^{-3}$ and $\theta$ (potential temperature referenced to the sea surface) $\leq 0$ °C[3,11]. Using data from a large number of shipboard surveys, an Arctic-origin overflow transport mode of the NIJ was identified, where the bulk of the transport is confined to a small region in potential temperature-salinity ($\theta$-$S$) space[5]. This mode is centered near $-0.27$ °C in potential temperature and 34.91 in practical salinity, corresponding to $\sigma_0 = 28.05$ kg m$^{-3}$ and accounting for 26% of the total overflow transport[5]. By comparison, at the Faroe Bank Channel sill the densest component ($\theta \leq 0$ °C) contributes to more than half of the total overflow transport[11]. This component has nearly the same properties ($\overline{\sigma_0} = 28.05$ kg m$^{-3}$, $\overline{\theta} = -0.4$ °C and $\overline{S} = 34.91$) as the transport mode of the NIJ. The similarity of the hydrographic properties of these two dense modes suggests a common source. In this study, we focus on this densest mode and reveal that it indeed emanates from a single geographical region: the Greenland Sea gyre. We then track the progression of the water towards the Greenland-Scotland Ridge where it ultimately overflows and ventilates the abyssal North Atlantic.

## Results and discussion

**A new metric to trace the source of the densest overflow**. We use a comprehensive historical hydrographic dataset of the Nordic Seas, described in the "Methods" section, to elucidate the source regions and pathways of the Arctic-origin overflow water feeding Denmark Strait and the Faroe Bank Channel. A recently-developed metric, referred to as the $\sigma_0$–$\pi_0$ distance, is used to calculate how similar two different water parcels are in terms of physical properties, where $\sigma_0$ is potential density and $\pi_0$ is potential spicity[12]. The metric is effective because the isolines of $\sigma_0$ are orthogonal to the isolines of $\pi_0$, and their gradients have the same magnitude (see the "Methods" section). Here the Arctic-origin overflow water is defined by the transport mode of the NIJ in $\sigma_0$–$\pi_0$ space, $\sigma_0 = 28.05$ kg m$^{-3}$ and $\pi_0 = -3.11$ kg m$^{-3}$ (the

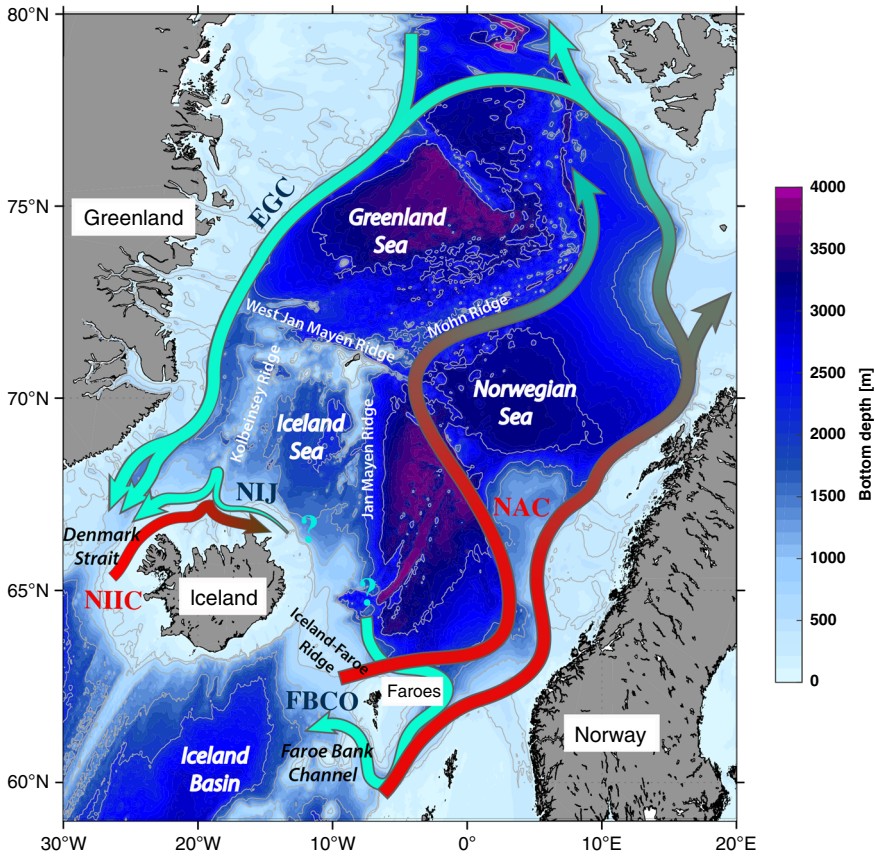

**Fig. 1 Schematic circulation of the Nordic Seas.** The pathways of warm Atlantic inflow and dense outflow are shown in red and green arrows, respectively. Colors and gray contours represent the bathymetry from ETOPO2, and the relevant ridges are named. NAC Norwegian Atlantic Current, NIIC North Icelandic Irminger Current, EGC East Greenland Current, NIJ North Icelandic Jet, FBCO Faroe Bank Channel overflow. Question marks denote uncertainty in the upstream source/pathways of the dense overflows.

results are nearly indistinguishable using the $\sigma_0-\pi_0$ values of the densest Faroe Bank Channel overflow water). Computationally, we determine the $\sigma_0-\pi_0$ distance between the water parcels in question upstream of the Greenland-Scotland Ridge and the NIJ transport mode. We consider the time period 2005-2015 to avoid effects of decadal variability, which are addressed in the last section of the paper.

We begin by asking, where in the Nordic Seas are the late-winter surface properties most closely aligned with the Arctic-origin overflow water? To answer this, we determined the mixed-layer characteristics (temperature, salinity, density, and depth) for approximately 8600 late-winter (February to April) profiles, using a multi-step procedure[13]. Not surprisingly, the deepest and densest mixed layers are found within the Greenland Sea gyre due to the strong atmospheric forcing and weak stratification there[14,15] (Supplementary Fig. 1). The Iceland Sea gyre also stands out as a region of somewhat deep, dense mixed layers, but to a lesser extent than the Greenland Sea. Notably, the smallest $\sigma_0-\pi_0$ distances associated with the mixed layers (as small as

0.005 kg m$^{-3}$) are found in the Greenland Sea (Fig. 2a), indicating that this is the major source region of the densest overflow water enting the North Atlantic.

To estimate the annual production of this dense water in the Greenland Sea gyre, we constructed composite vertical sections across the center of the gyre for the autumn time period (before convection) and late-winter time period (after convection). Assuming a gyre radius of 150 km, the change in volume of the small $\sigma_0-\pi_0$ distance water (distance < 0.05 kg m$^{-3}$, mean $\sigma_0 = 28.05$ kg m$^{-3}$) between the two periods is $1.8 \pm 0.4 \times 10^4$ km$^3$. This translates to yearly formation rate of $0.6 \pm 0.1$ Sv, where the uncertainty represents the standard error. We note that this estimated production is biased low because of the unaccounted export of dense water during winter. However, it is comparable with the annual transport of the densest overflow in the NIJ as well that flowing across the sill at Denmark Strait[3,16] (0.5–0.6 Sv).

A second region of relatively small $\sigma_0-\pi_0$ distances occurs in the northwest portion of the Iceland Sea (Fig. 2a), consistent with the notion that this region can supply some of the Arctic-origin

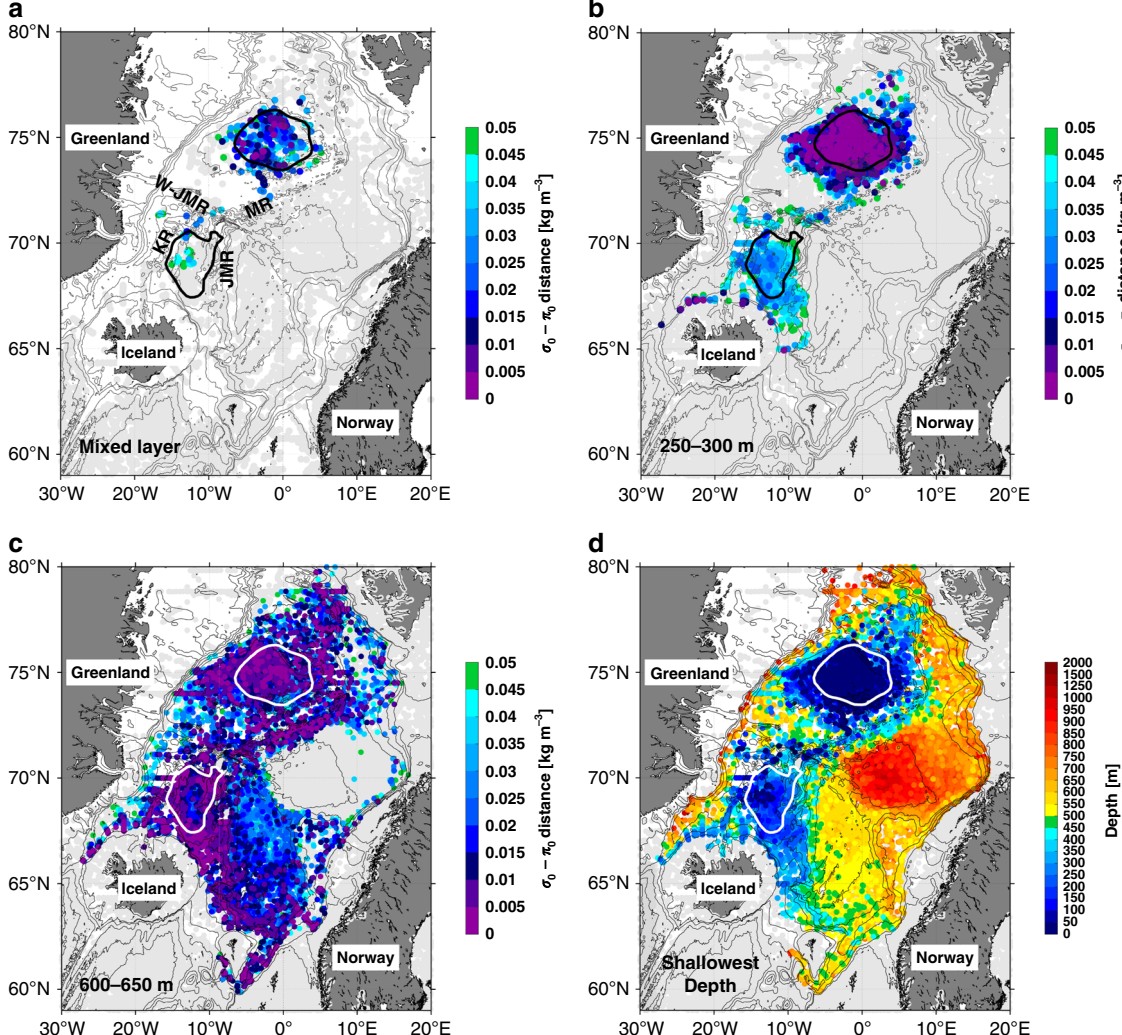

**Fig. 2 Proximity in water mass space to the densest overflow water. a** Distribution of the small potential density–potential spicity ($\sigma_0$-$\pi_0$) distances (≤0.05 kg m$^{-3}$) for late-winter (February–April) mixed layers from 2005 to 2015. Distances >0.05 kg m$^{-3}$ are shown by light-gray circles. The Greenland Sea and Iceland Sea gyres are delimited by the 4.5 and 7 dynamic cm contours, respectively, of sea surface dynamic height relative to 500 db (thick black and white contours). The thin gray contours show the bathymetry from ETOPO2, with four ridges labeled: MR, Mohn Ridge; W-JMR, West Jan Mayen Ridge; JMR, Jan Mayen Ridge; KR, Kolbeinsey Ridge. **b**, **c** The year-round distribution of small $\sigma_0$-$\pi_0$ distances in the 250–300 m layer and 600–650 m layer, from 2005 to 2015. **d** The shallowest depth at which a small $\sigma_0$-$\pi_0$ distance value (0.05 kg m$^{-3}$) is first found in the vertical. Stations where this value is not attained are shown by light-gray circles.

overflow water to the NIJ[17,18]. However, only a small number of hydrographic profiles (5%) recorded small $\sigma_0-\pi_0$ distances (<0.05 kg m$^{-3}$) in the late-winter mixed layers in the Iceland Sea. Furthermore, the mean density of these profiles is $\sigma_0 = 28.01$ kg m$^{-3}$ which is significantly less dense than the small $\sigma_0-\pi_0$ distance water formed in the Greenland Sea, and the winter mixed layers in the Iceland Sea only extend to a few hundred meters (Supplementary Fig. 1). Our results thus imply that the Iceland Sea plays a relatively minor role in the production of the densest overflow water feeding the North Atlantic.

To investigate the connection between the formation region of the Arctic-origin overflow and the presence of this water in Denmark Strait and the Faroe Bank Channel, we calculated the $\sigma_0-\pi_0$ distances at different depth horizons in the Nordic Seas using the year-round hydrographic profiles. This reveals where the water is present after late-winter subduction. Fig. 2b shows the $\sigma_0-\pi_0$ distance distribution between 250 and 300 m, which is the depth range of the maximum velocity in the NIJ[19]. The smallest $\sigma_0-\pi_0$ distances still occur in the Greenland Sea. However, a potential pathway of the Arctic-origin overflow water emerges from this map. In particular, small $\sigma_0-\pi_0$ distances stem southward from the Greenland Sea along the Mohn Ridge, across the West Jan Mayen Ridge into the Iceland Sea, then along the Kolbeinsey Ridge—ultimately progressing into Denmark Strait via the NIJ. The consistently small $\sigma_0-\pi_0$ distances from the Greenland Sea to Denmark Strait demonstrate that the water mass properties do not change substantially along the pathway. Note that no small $\sigma_0-\pi_0$ distances are found along the continental slope of Greenland, confirming that Arctic-origin overflow water is not advected by the East Greenland Current in this depth range.

The lateral distribution of $\sigma_0-\pi_0$ distances in the 600–650 m layer, near the sill depth of Denmark Strait, is generally consistent with the shallower layer (Fig. 2c). However, there are two notable differences. First, small $\sigma_0-\pi_0$ distances now extend eastward along the northern slope of the Iceland–Faroe Ridge into the Faroe Bank Channel. Second, there are two bands of small values on either side of the Iceland Sea: one along the eastern side of the Kolbeinsey Ridge (similar to that seen in the shallower layer), and the other along the Jan Mayen Ridge. This suggests that, at this deeper depth horizon, two potential pathways exist carrying Arctic-origin overflow water from the Greenland Sea to the Faroe Bank Channel along the north-south ridges bounding the Iceland Sea (recall that local ventilation in the Iceland Sea does not reach this deep). The $\sigma_0-\pi_0$ distance map for the depth range of the Faroe Bank Channel sill (800–850 m) is qualitatively similar to that in Fig. 2c. When using the Faroe Bank Channel transport mode instead of the NIJ transport mode to compute the $\sigma_0-\pi_0$ distances, the Jan Mayen Ridge pathway is even more evident in this depth range (not shown).

As a metric of the vertical structure of the $\sigma_0-\pi_0$ distances, in Fig. 2d we plot the shallowest depth at which the value equals 0.05 kg m$^{-3}$ (which is the threshold used in Fig. 2a–c to define small $\sigma_0-\pi_0$ distances). This map also reveals the pathways discussed above for Arctic-origin water to progress from the Greenland Sea gyre to the two overflows. Interestingly, it implies as well that the small $\sigma_0-\pi_0$ distance signal stays at roughly the same depth approaching Denmark Strait in the NIJ, but that it deepens approaching the Faroe Bank Channel along the Iceland–Faroe Ridge. Also in Fig. 2d one sees that the shallowest depth of small $\sigma_0-\pi_0$ distances along the continental slope of East Greenland is deeper than the sill depth of Denmark Strait (650 m). This implies that the Arctic-origin water located below the Atlantic-origin water in the EGC is too deep to contribute to the densest Denmark Strait overflow, in contrast to previous assertions[20,21]. However, there is evidence of aspiration of

Arctic-origin water just upstream of the sill[16], which may reconcile these seemingly contrasting views.

Previous studies have shown a very weak seasonal signal in the overflows at Denmark Strait and Faroe Bank Channel[11,22]. Here we investigated the seasonality of the densest overflow water by constructing timeseries of the layer thickness of small $\sigma_0-\pi_0$ distance water (< 0.05 kg m$^{-3}$) in the upper 650 m, i.e., above the sill depth of Denmark Strait (results are indistinguishable using the Faroe Bank Channel sill depth of 850 m). Four regions were chosen corresponding to the progression of the water emanating from the Greenland Sea gyre, as described above (Supplementary Fig. 2a). In the gyre one clearly sees the wintertime formation of the water (Supplementary Fig. 2b), while along the Mohn Ridge the newly ventilated water appears in spring/early-summer (Supplementary Fig. 2c). However, there is no evidence of a seasonal signal along the two pathways bounding the Iceland Sea (Supplementary Fig. 2d–e). One possible explanation is hydraulics, which tends to suppress time variability[23]. As the dense water crosses the West Jan Mayen Ridge into the Iceland Sea it presumably does so through gaps in the ridge. Waves carrying information about upstream time dependence will not get completely transmitted across the ridge if the gaps are hydraulically controlled[24]. This and other potential causes for the disappearance of the seasonality of the densest overflow water away from the source region requires further investigation.

**Upstream overflow pathways derived from composite sections.** The distributions of $\sigma_0-\pi_0$ distances presented above suggest that the Arctic-origin overflow water crossing the Greenland–Scotland Ridge at Denmark Strait and the Faroe Bank Channel originates primarily from the Greenland Sea, and that there are topographically-steered pathways by which the water progresses from the Greenland Sea gyre to the vicinity of the ridge. We now present kinematic support of these pathways using the historical hydrographic data in conjunction with satellite absolute dynamic topography data. We focus on five composite sections (Fig. 3a) that cross the ridge systems as well as the Iceland and Greenland Sea gyres, chosen to elucidate the circulation in question.

Relative geostrophic velocities were computed using the hydrographic data, then referenced using the mean gridded surface geostrophic velocity field obtained from satellite altimeter data (see the "Methods" section). The surface geostrophic vectors show the major currents in the study region (Fig. 3b). The two branches of the Norwegian Atlantic Current (NAC) flow northward, with the western branch turning to the northeast along the Mohn Ridge on the eastern side of the Greenland Sea gyre. The southward-flowing EGC is evident, as is the North Icelandic Irminger Current (NIIC) flowing around the north side of Iceland. There is relatively little surface signature of the Greenland Sea gyre nor the Iceland Sea gyre.

The first two composite sections (S1 and S2) reveal the cold near-surface water in the center of the Greenland Sea gyre (Fig. 3c, d). The strongly sloped isopycnals on the western side are associated with the EGC, while those on the eastern side are associated with the NAC. The flat isopycnals between the two fronts are consistent with the weak surface flow of the gyre. The third composite section (S3) is south of the Greenland Sea gyre, but still shows the strong EGC and NAC fronts. The last two composite sections (S4 and S5) pass through the northern and central portions of the Iceland Sea gyre, respectively. Here the NAC front is not associated with the eastern edge of the gyre, but is found farther to the east. However, the EGC abuts the western side of the gyre, as it did for the Greenland Sea. Overall, the fronts in the hydrographic sections match well with the major currents

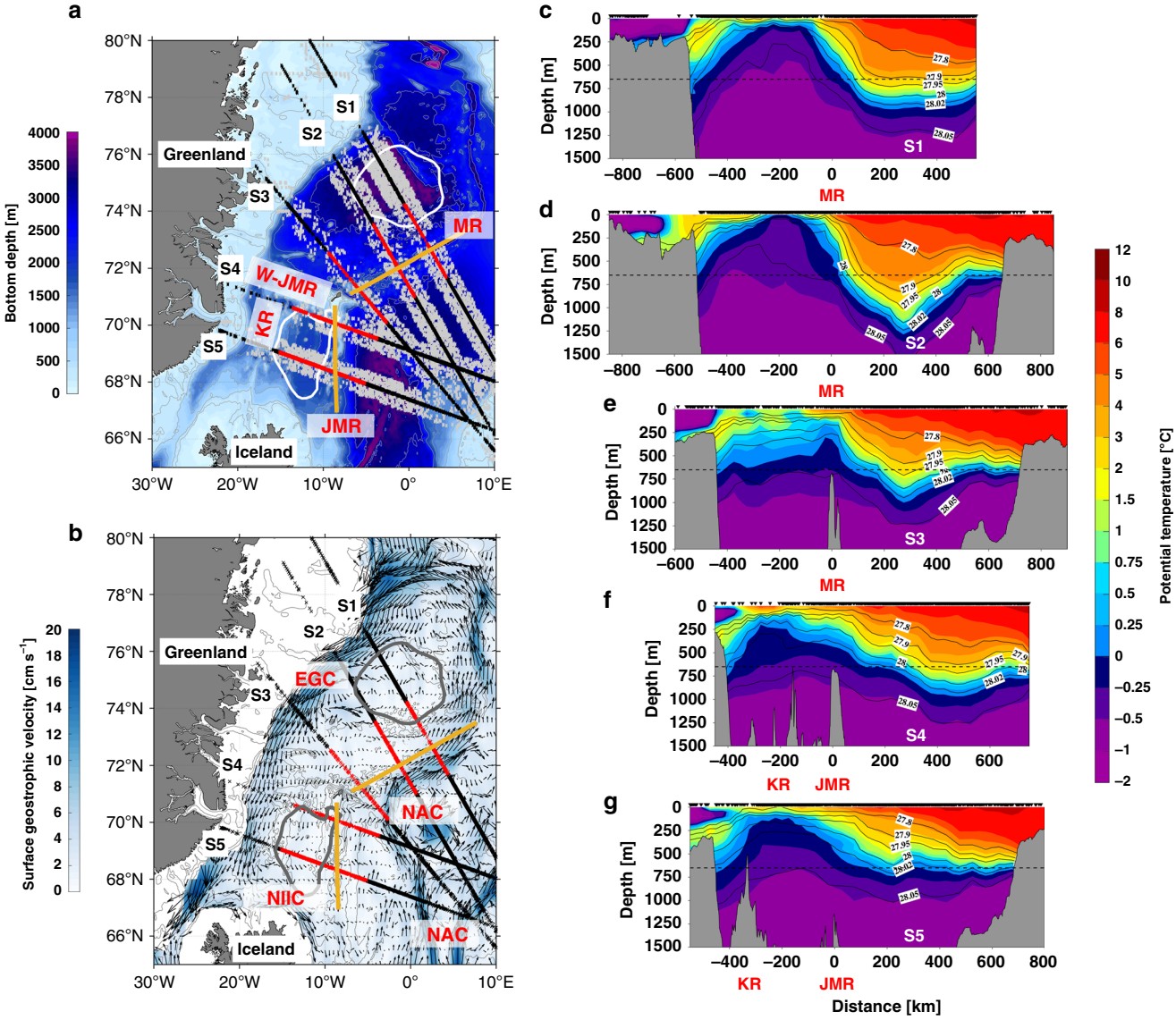

**Fig. 3 Cross-pathway sections of potential temperature and potential density. a** Locations of the five composite sections (S1–S5) centered on the Mohn Ridge (MR) and Jan Mayen Ridge (JMR) (yellow lines, see Fig. 2 for all ridge names). The original and projected locations of hydrographic profiles used in each section are indicated by gray and black crosses, respectively. The red crosses denote the segment of the section used for calculating the absolute geostrophic velocity in Fig. 4. The Greenland Sea and Iceland Sea gyres are outlined by the white contours. **b** The long-term mean (2005-2015) surface geostrophic velocity from the gridded satellite altimeter product. The two branches of the Norwegian Atlantic Current (NAC), the East Greenland Current (EGC), and the North Icelandic Irminger Current (NIIC) are labeled. The color indicates speed [cm s⁻¹]. **c–g** Cross-sections of potential temperature [°C] in color, overlain by potential density contours [kg m⁻³]. The origin of the x axis in S1-S3 and in S4-S5 are the locations of the Mohn Ridge and Jan Mayen Ridge, respectively. The horizontal dashed back line indicates the sill depth of Denmark Strait (650 m).

in the surface geographic velocity field (the composite salinity sections are shown in Supplementary Fig. 3).

We present the calculated circulation at three depth horizons: the surface, 300 m and 650 m (Fig. 4). The latter two levels correspond to the $\sigma_0-\pi_0$ distance maps in Fig. 2b, c, respectively. At the surface (Fig. 4a) the warm NAC flows northward east of the Mohn Ridge (as was noted earlier in Fig. 3b–d). The hydrographic front of the NAC is associated with strong thermal wind shear, which is large enough to reverse the circulation at 300 m and deeper (sections S2 and S3 in Fig. 4b, c). This provides a pathway of Arctic-origin overflow water away from the Greenland Sea gyre that continues southward on the eastern side of the Iceland Sea along the Jan Mayen Ridge (sections S4 and S5 in Fig. 4b, c). The circulation maps also reveal a second dense water pathway that emerges at section S2 west of the Mohn Ridge

and strengthens by section S3. This branch passes through the West Jan Mayen Ridge and flows southward in the Iceland Sea on the eastern side of the Kolbeinsey Ridge (sections S4 and S5, Fig. 4b, c). Overall, these circulation pathways are consistent with the $\sigma_0-\pi_0$ distance distributions presented above, including the two bands of low $\sigma_0-\pi_0$ distance on the two sides of the Iceland Sea (Fig. 2c).

The two branches of Arctic-origin overflow water emanating from the vicinity of the Mohn Ridge appear to be dynamically distinct. The western branch is nearly barotropic, with a signature at the sea surface starting at section S3 (Fig. 4a and Supplementary Fig. 4). In contrast, the eastern branch is sub-surface intensified with stronger flow at 650 m (Fig. 4c and Supplementary Fig. 4). The fate of these two branches remains to be determined. Previous studies give differing views regarding the

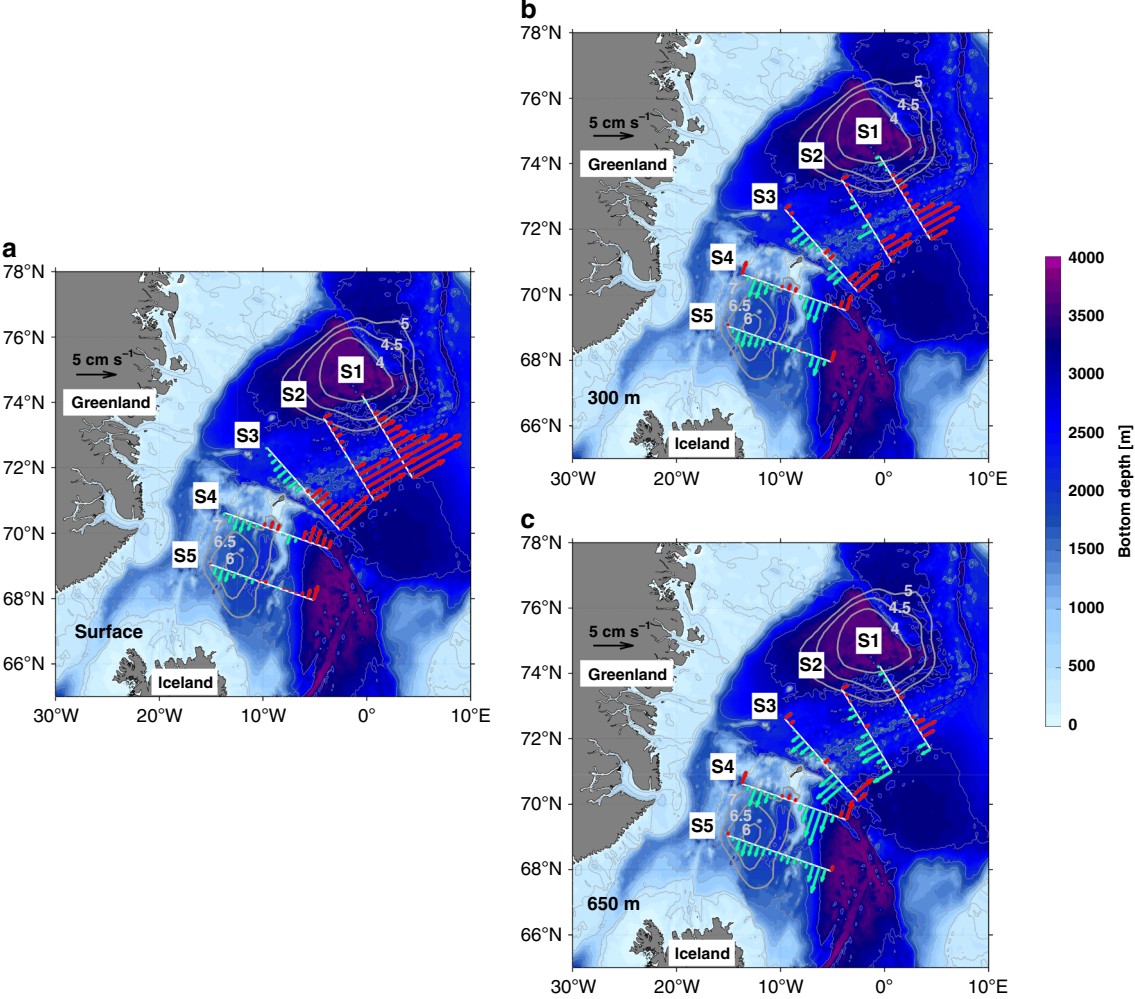

**Fig. 4 Velocities along the pathways.** Absolute geostrophic velocity [cm s$^{-1}$] normal to the five composite sections for (**a**) the surface, (**b**) 300 m, and (**c**) 650 m. The southward and northward flows in each section are shown in blue and red, respectively. The Greenland Sea and Iceland Sea gyres are indicated by contours of sea surface dynamic height relative to 500 db in units of dynamic cm (thick gray contours).

origin of the NIJ. Model simulations have suggested both the central Iceland Sea[3] and the southern Iceland Sea[25] as source regions, while Lagrangian measurements imply an eastern source[26]. With regard to the Faroe Bank Channel overflow, a recent study has revealed a bottom intensified current of dense overflow water progressing eastward along the north side of the Iceland–Faroe Ridge, presumably supplying the overflow[27]. It is likely that the two branches of Arctic-origin overflow water identified here feed both the NIJ and this current, but the precise partitioning and manner in which this happens needs to be worked out. Notably, the suggested pathways presented here are consistent with a two-layer modeling study[28] and a tracer release experiment[29].

**Long-term variation in the source of the densest overflow.** We now consider the complete time period of the historical hydrographic observations to address long-term differences in the source of the Arctic-origin overflow water to the Greenland-Scotland Ridge. Since there is not enough data coverage throughout the Nordic Seas to resolve interannual variability, we consider a single early period, 1986–2004, and compare this to the 2005–2015 results presented above (the total amount of data is comparable in both periods). While the temperature and salinity

of the NIJ water have varied from 1990 to the present[18], the magnitude of these variations in $\sigma_0-\pi_0$ space is small (0.01–0.02 kg m$^{-3}$) compared with the threshold of small $\sigma_0-\pi_0$ distance (0.05 kg m$^{-3}$). Therefore, we use the same NIJ transport mode to calculate the $\sigma_0-\pi_0$ distances for the early period.

The primary difference in source water between the two time periods is most effectively shown by considering the relative layer thickness of the small $\sigma_0-\pi_0$ distance water in the upper 650 m of the water column (above the sill depth of Denmark Strait; results are indistinguishable using the Faroe Bank Channel sill depth). This was done via the following steps. First, profiles of $\sigma_0-\pi_0$ distance were computed between each hydrographic profile over the domain and the NIJ transport mode. Next, for each profile we calculated the percent thickness of the layer where the $\sigma_0-\pi_0$ distance was <0.05 kg m$^{-3}$, relative to the 650 m depth range. Finally, we bin-averaged the results using a grid of 1° longitude by 1/3° latitude.

The resulting percentages are shown in Fig. 5a, b, which reveals a marked increase in the presence of Arctic-origin overflow water (i.e., small $\sigma_0-\pi_0$ distance water) in both the Greenland and Iceland Seas in the later time period. In addition, the layer thickness of small $\sigma_0-\pi_0$ distances outside the Greenland Sea gyre (166 ± 172 m) is higher than that inside the Greenland Sea gyre (131 ± 154 m) in the earlier time period (see the preponderance

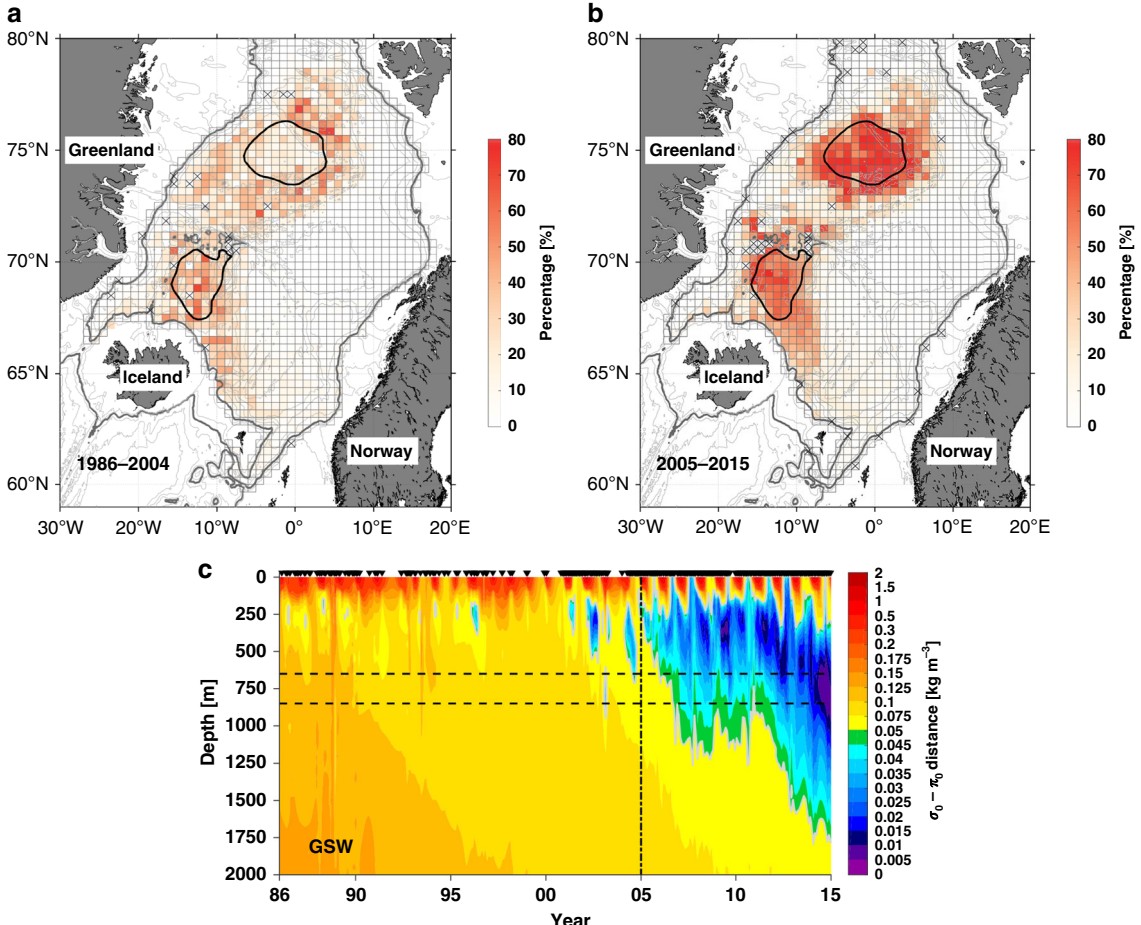

**Fig. 5 Long-term variability of the densest overflow water.** Relative layer thickness (in percentage, colors) of the small $\sigma_0$–$\pi_0$ distance water ($\leq$0.05 kg m$^{-3}$) in the upper 650 m of Nordic Seas for the time period (**a**) 1986–2004, and (**b**) 2005–2015. The thick gray contours show the 650 m isobath. The bins marked with an X denote the regions with no data coverage. **c** Evolution of $\sigma_0$–$\pi_0$ distances between the NIJ transport mode and the water within upper 2000 m of the Greenland Sea gyre (the black contour in **a** and **b**). The two horizontal dashed lines indicate the sill depths of Denmark Strait (650 m) and the Faroe Bank Channel (850 m). The vertical dashed line denotes year 2005. The gray contours denote $\sigma_0$–$\pi_0$ distances = 0.05 kg m$^{-3}$.

of red color surrounding the Greenland Sea gyre in Fig. 5a). The uncertainty represents the standard deviation. This result indicates a shift in the origin of the densest water feeding the overflows, from the periphery of the Greenland Sea gyre during the earlier period, to its center during the later period. During both time periods, however, the pathways of small $\sigma_0$–$\pi_0$ distances remained the same (not shown): from the Mohn Ridge into the Iceland Sea, then along the Jan Mayen and Kolbeinsey Ridges on the eastern and western sides of the Iceland Sea, respectively.

Why was the interior of the Greenland Sea gyre not the primary dense water source during the early time period? This is due to the change in convection within the gyre over the 30-year record, which can be assessed on a yearly basis due to the good data coverage within the gyre. The vertical distribution of $\sigma_0$–$\pi_0$ distance averaged over the gyre shows values <0.05 kg m$^{-3}$ spanning the top 650 m starting around 2005 (Fig. 5c, with small values extending to the surface seasonally in winter). This is associated with warming and salinification of the ventilated water in the gyre due to weakening of deep convection and changes to the inflowing Atlantic water into the Nordic Seas[15,30,31]. Prior to 2005, the dense wintertime product in the center of the gyre was too cold and fresh to match closely the properties of the two overflow transport modes. Such a change in source region between the two periods, from outside the gyre to within the gyre, likely has ramifications regarding the dynamical processes and

timescale for newly ventilated water to progress from the formation region to the pathways supplying the overflows.

In this study, we have elucidated the source and upstream pathways of the densest Arctic-origin water supplying the Denmark Strait and Faroe Bank Channel overflows, using a new metric for determining how similar two water parcels are in terms of hydrographic properties. Kinematic evidence for the inferred pathways was provided via absolutely-referenced circulation maps. Our results establish the Greenland Sea as the primary region of ventilation for the deepest layers of the North Atlantic, and reveal the importance of topographically steered pathways in channeling the dense water towards the Greenland-Scotland Ridge. This puts us in a position to better understand the link between atmospheric forcing and the densest component of the AMOC, and how this might be impacted by ongoing changes in the climate of the high latitude North Atlantic. Our study also raises several new questions, including: How and why does the newly-convected water enter the deep Mohn Ridge boundary current system after it is subducted? And how does this dense water subsequently pass through the West Jan Mayen Ridge into the Iceland Sea? Finally, it remains to be determined how the dense Arctic-origin water ultimately feeds the westward-flowing NIJ and eastward-flowing current along the Iceland–Faroe Ridge. These questions need to be addressed with future observational and modeling studies.

## Methods

**Hydrographic data**. The hydrographic data used in this study cover the time period 1986–2015 and the spatial domain 59°–80°N and 30°W–20°E. The majority of the data were obtained from the Unified Database for Arctic and Subarctic Hydrography (UDASH)[32]. Additional data, particularly south of 65°N which is outside the domain of UDASH, come from various archives as listed in supplementary Table 1. All of the data were combined into a single hydrographic dataset, where duplicates between the different archives were removed. In addition to the quality control already performed on each data source, we required that all vertical profiles include both temperature and salinity. We excluded measurements outside the expected range in the Nordic Seas [−2 °C, 20 °C] and [20, 36] for potential temperature and practical salinity, respectively. Data with density inversions exceeding 0.05 kg m$^{-3}$ were also excluded, except when the inversion was associated with a single data point, in which case the point in question was removed[15].

**$\sigma_0$–$\pi_0$ distance metric**. A metric known as the $\sigma_0$–$\pi_0$ distance is used to calculate how similar two different water parcels are in terms of physical properties, referenced to the sea surface, where $\sigma_0$ is potential density and $\pi_0$ is potential spicity[12]. This metric has been successfully applied to classify water masses in the South China Sea[33]. Contours of $\pi_0$ are orthogonal to those of $\sigma_0$, hence $\pi_0$ contains information regarding temperature and salinity not included in the potential density. In contrast to $\theta$–$S$ space, the gradients in $\sigma_0$–$\pi_0$ space are the same magnitude, allowing a meaningful calculation of distance. The $\sigma_0$–$\pi_0$ distance between the densest overflow water ($\sigma_{0,1}$, $\pi_{0,1}$) and any upstream water parcel ($\sigma_{0,2}$, $\pi_{0,2}$) in the Nordic Seas is calculated by the following equation:

$$D_{1,2} = \sqrt{(\sigma_{0,1} - \sigma_{0,2})^2 + (\pi_{0,1} - \pi_{0,2})^2} \qquad (1)$$

**Composite sections and absolute geostrophic velocities**. The composite vertical sections were constructed using profiles from the historical hydrographic dataset as follows. For each selected section, profiles with lateral distances less than 50 km from the line are used (gray crosses in Fig. 3a). The positions of profiles along the section are determined by the distance between their projected location (black crosses in Fig. 3a) and the location of the ridge (which is the origin of the section). The projected profiles are then gridded using Laplacian-spline interpolation[34]. The final gridded sections have a horizontal resolution of 25 km and a vertical resolution of 25 m. The relative geostrophic velocities normal to the sections are calculated based on the gridded data. The absolute geostrophic velocities are obtained by using satellite-derived mean surface geostrophic velocity from 2005 to 2015 as the reference, which can be accessed at Copernicus Marine Environment Monitoring Service (CMEMS, http://marine.copernicus.eu).

## Data availability

Access to the individual hydrographic data sources is listed in Supplementary Table 1. The combined hydrographic dataset is available on request from A.B. (Ailin. Brakstad@uib.no). The surface geostrophic velocities were obtained from Copernicus Marine Environment Monitoring Service (CMEMS, http://marine.copernicus.eu).

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

## Acknowledgements

Funding for the study was provided by the US National Science Foundation under grants OCE-1558742 (J.H., R.P.) and OCE-1259618 (P.L.); the Bergen Research Foundation under grant BFS2016REK01 (A.B.); and the National Natural Science Foundation of China No. 41576018 (F.X.) and 41606020 (F.X.).

## Author contributions

J.H., R.P., and A.B. assembled and analyzed the data; R.H. developed the $\sigma_0$–$\pi_0$ distance methodology and assisted in its application; J.H. and R.P. wrote the paper; J.H., R.P., R.H., P.L., A.B., and F.X. interpreted the results and clarified the implications.

## Competing interests

The authors declare no competing interests.
