## [Peer Review File · Nature Communications]

REVIEWER COMMENTS

Reviewer #1 (Remarks to the Author):

This paper uses hydrographic measurements, over recent decades, and satellite observation of sea surface height to investigate the source regions of the densest overflows from the Nordic Seas to the subpolar North Atlantic. The authors facilitate a method to quantify how similar two water parcels are, based on potential density and potential spiciness. In this particular case they compute this quantity relative to the densest water in the Denmark Strait and Faroe Bank Channel. Their results show that the closest similarity to the overflows is obtained with water parcels in the Greenland Sea, which they conclude is the main formation region for the overflows. Their diagnostics suggest that the Greenland Sea has become a larger contributor to the overflow in the recent decade.

This paper covers an interesting topic and applies a new metric to obtain new insights where the dense water is formed. The results are compelling and support their argumentation. A few questions arose while reading through the manuscript, which the authors might like to address. At present the metric measures how similar water parcels are and certainly the Greenland Sea shows the largest agreement with the overflow water. However, this metric does not provide a quantitative measure how much water is formed with the same properties and therefore saying that the Greenland Sea is the main source region, needs to be treated with caution.

In Figure 2c suggest that properties in Greenland and Iceland Sea agree with overflow properties and a dominance of the Greenland Sea is not obvious. Note that the chosen projection makes the Greenland Sea larger than it is. Do the authors suggest that water formed in the Greenland Sea 'invades' the Iceland Sea and the Iceland Sea does not play a role? What does the 'hole' in the centre of the Iceland Sea tell us in Figure 2c. Does wintertime convection produce an even denser water mass? At the moment the author assume that water mass properties do not change along the path towards the overflows.

If overflow water is generated as described in the Greenland Sea by wintertime convection, would not we expect a seasonal signal to be present in the overflows. To my knowledge there is only a very weak seasonal signal present in observations. How does that fit? Is it due to the distance from the source region?

I do not get much value out of the Figure 3c-g based on the provided description.

Figure 5. The overflows do not show any substantial long-term trends. So, the question is if the formation region in 1986-2004 was not where it is in the recent decades, where was it? The current figure would suggest there was less of it in the earlier period, by looking how much red there is.

Reviewer #2 (Remarks to the Author):

This manuscript by Jie Huang et al. presents an analysis on the source and pathways of the densest component of the overflow waters from the Nordic Seas using a recently-developed metric of potential density - potential spicity distances. The manuscript is very well written, clearly structured with appropriate figures and the results are novel. I only have a few comments and therefore would recommend minor revisions before publication in Nature Communications.

Major comments

This study provides a very clear and thorough analysis of the sources and pathways of the densest component of the overflows. While reading, there were two aspects that I feel are missing in the discussion of the results.

First, a discussion is missing regarding pathway and source variability related to previous findings of e.g. Köhl 2010 (<https://www.tandfonline.com/doi/abs/10.1111/j.1600-0870.2009.00454.x>), de

Jong et al. 2018 (<https://doi.org/10.1016/j.dsr.2018.07.008>) and Mauritzen 1996 ([https://doi.org/10.1016/0967-0637\(96\)00038-6](https://doi.org/10.1016/0967-0637(96)00038-6)). Previously, the Greenland Sea gyre was assumed to play a minor role for the overflows as the variability of open ocean convection was very large compared to the transport variability of the overflows. In this study, the focus is specifically on the Arctic-origin part of the overflow, considered to be mostly carried by the NIJ. How does the variability of this watermass at Denmark Strait and FBC compare to the dense water production in the Greenland Basin? Previous studies suggest that the strength of the NIJ is very variable and that export of the Greenland Sea watermass along the EGC is also a possibility (Köhl 2010, Messias et al. (<https://doi.org/10.1016/j.pocean.2007.06.005>)). How does this compare to your findings, as the EGC pathway is not found? Does the EGC pathway only exist for less dense waters of the Greenland Basin? Or is there still a possibility for the densest waters to leave the Greenland Basin along the EGC, but are they substantially modified along the way and therefore no longer contribute the densest part of the overflow? Some elaboration on these aspects would strengthen the manuscript.

Second, a discussion is missing on how the findings compare to findings from Lagrangian observation and modelling studies regarding the origin of the overflows (e.g. Behrens et al. 2017 (<https://doi.org/10.1002/2016JC012158>)). Ln. 168 is a bit too limited in that respect.

Minor comments

Ln. 22 and throughout the manuscript: For clarity I would rephrase 'how close two water parcels are to each other' to 'how similar two water parcels are in terms of physical properties'.

Ln. 24 and Ln 29: At reading only the abstract, it is not directly clear what is meant with the Greenland Sea gyre versus the periphery of the Greenland Sea gyre. Maybe replace 'gyre' with 'basin' or 'interior'?

Ln. 56: The ending of the introduction is a bit abrupt. Regarding the last sentence it seems as if the main aim of the paper is to investigate whether the densest component of the FBC overflow has a similar origin as the NIJ water mass. I would advice to make more clear what the main aim is of the paper and how the paper is structured.

Ln. 84: Can you explain what exactly you mean with 'a minor source' and how you conclude that from Fig. 2a? Number of measurements or magnitude of potential density - potential spicity distance does not necessarily imply how much source A contributes to the overflow and how much source B contributes in my understanding.

Ln. 124: I am not sure whether 'evidence' is the right term here. The Eulerian velocity fields indeed show that the pathways suggested by the potential density - potential spicity distance analysis are likely, but a combination with a Lagrangian approach is needed to really prove the connectivity.

Ln. 127: to resolve → to investigate? With 5 transects you're not resolving the circulation.

Ln. 174: Is the spatial and temporal distribution of the observations used for analysis in the early period comparable to the later period?

Ln. 176: Are these variations small compared to the maximum potential density - potential spicity distances used to find the source regions? Please specify.

Ln. 213: to better understand

Figures in general: Could you increase the font size of text and colorbar labels etc.? At the moment it is barely readable on a A4 printout.

Fig. 2: It is difficult to distinguish the gray isobaths from the gray measurement circles, maybe use a different color?

Fig. 3a-b: The red text is very difficult to read.

We appreciate the reviewers' careful and constructive comments on the manuscript. Our responses are listed as follows in blue.

Reviewer #1 (Remarks to the Author):

This paper uses hydrographic measurements, over recent decades, and satellite observation of sea surface height to investigate the source regions of the densest overflows from the Nordic Seas to the subpolar North Atlantic. The authors facilitate a method to quantify how similar two water parcels are, based on potential density and potential spiciness. In this particular case they compute this quantity relative to the densest water in the Denmark Strait and Faroe Bank Channel. Their results show that the closest similarity to the overflows is obtained with water parcels in the Greenland Sea, which they conclude is the main formation region for the overflows. Their diagnostics suggest that the Greenland Sea has become a larger contributor to the overflow in the recent decade. This paper covers an interesting topic and applies a new metric to obtain new insights where the dense water is formed. The results are compelling and support their argumentation. A few questions arose while reading through the manuscript, which the authors might like to address.

At present the metric measures how similar water parcels are and certainly the Greenland Sea shows the largest agreement with the overflow water. However, this metric does not provide a quantitative measure how much water is formed with the same properties and therefore saying that the Greenland Sea is the main source region, needs to be treated with caution.

This is a good point, and we're glad that you brought it up. In the revised manuscript we now estimate the rate of production of small σ_0 - π_0 distance water in the Greenland Sea Gyre. In particular, we constructed composite vertical sections across the center of the gyre for the autumn time period (before convection) and late-winter time period (after convection). Assuming a gyre

radius of 150 km, the change in volume of the small σ_0 - π_0 distance water ($< 0.05 \text{ kg m}^{-3}$) between the two periods is $1.8 \pm 0.4 \times 10^4 \text{ km}^3$. This translates to a yearly formation rate of $0.6 \pm 0.1 \text{ Sv}$. We note that this estimated production is biased low because of the unaccounted export of dense water during the wintertime. However, it is comparable with the annual transport of the densest overflow in the NIJ as well that flowing across the sill at Denmark Strait (0.5 - 0.6 Sv , *Våge et al.*, 2011 and *Harden et al.*, 2016).

We also point out in the revised manuscript that only a small number of hydrographic profiles (5%) recorded small σ_0 - π_0 distances ($< 0.05 \text{ kg/m}^3$) in the late-winter mixed layers in the Iceland Sea. Furthermore, most of the overflow water ventilated in the Iceland Sea is less dense than that formed in the Greenland Sea, and the winter mixed layers in the Iceland Sea only extend to a few hundred meters (as demonstrated in Supplementary Fig. 1). These results together imply that the Iceland Sea plays a relatively minor role in the production of the densest overflow water feeding the North Atlantic. This is all included in the revised manuscript.

In Figure 2c suggest that properties in Greenland and Iceland Sea agree with overflow properties and a dominance of the Greenland Sea is not obvious. Note that the chosen projection makes the Greenland Sea larger than it is. Do the authors suggest that water formed in the Greenland Sea ‘invades’ the Iceland Sea and the Iceland Sea does not play a role? What does the ‘hole’ in the centre of the Iceland Sea tell us in Figure 2c. Does wintertime convection produce an even denser water mass? At the moment the author assume that water mass properties do not change along the path towards the overflows.

The reviewer is correct in that our data indicate that the Arctic-origin water formed in the Greenland Sea “invades” the Iceland Sea, and that ventilation in the Iceland Sea does not produce a substantial amount of the densest overflow water. As stated above, we now clarify in the revised manuscript that the winter mixed layers in the Iceland Sea only extend a few hundred meters and

that they are mostly less dense than the densest overflow water. Hence, the depth level of Fig. 2c is not indicative of local formation in the Iceland Sea. Rather, it is indicative of formation in the Greenland Sea together with the pathways emanating from the Greenland Sea. This is why the Greenland Sea gyre does not appear dominant at this depth level compared to the shallower depth level of Fig. 2b that more effectively isolates the overflow water within the gyre.

Regarding the “hole” in the Iceland Sea, we have demonstrated that there are two pathways by which the dense water formed in the Greenland Sea passes through the Iceland sea – along the two north-south ridges that bracket the Iceland Sea gyre. This explains the hole in the center of the Iceland Sea gyre. We have now made this more clear in the revision.

We don't need to assume that the water properties do not change substantially along the pathways towards the overflow. The consistently small σ_0 - π_0 distances along the pathways demonstrate this. We now state this explicitly in the revision.

If overflow water is generated as described in the Greenland Sea by wintertime convection, would not we expect a seasonal signal to be present in the overflows. To my knowledge there is only a very weak seasonal signal present in observations. How does that fit? Is it due to the distance from the source region?

This is a good point. In the revised manuscript, we now discuss the seasonality of the densest Arctic-origin overflow water and include a new supplementary figure (Fig. S2). In particular, we constructed timeseries of the layer thickness of small σ_0 - π_0 distance water ($< 0.05 \text{ kg m}^{-3}$) in the upper 650 m, i.e. above the sill depth of Denmark Strait (results are indistinguishable using the Faroe Bank Channel sill depth of 850m). Four regions were chosen corresponding to the progression of the water emanating from the Greenland Sea gyre, as described earlier in the paper.

In the gyre one clearly sees the wintertime formation of the water, while along the Mohn Ridge the newly ventilated water appears in spring/early-summer. However, there is no evidence of a seasonal signal along the two pathways bounding the Iceland Sea. We invoke hydraulics as a possible explanation, and include references. We note that as the dense water crosses the West Jan Mayen Ridge into the Iceland Sea, it presumably does so through gaps in the ridge. Waves carrying information about upstream time dependence will not get completely transmitted across the ridge if the gaps are hydraulically controlled. A detailed investigation of this, however, is beyond the scope of our study.

I do not get much value out of the Figure 3c-g based on the provided description.

Admittedly we don't discuss Fig. 3c-g much in the paper, but we feel that it provides important context to our story. Namely, it shows the reader the isopycnal doming of the two gyres and the large presence of the Arctic-origin overflow water in the Greenland Sea gyre. It also shows the reader why the surface dynamic height signature of both gyres is weak. We prefer to keep the figure in the paper. We now point out in the revised manuscript that the fronts in the hydrographic sections match well with the major currents in the surface geographic velocity field.

Figure 5. The overflows do not show any substantial long-term trends. So, the question is if the formation region in 1986-2004 was not where it is in the recent decades, where was it? The current figure would suggest there was less of it in the earlier period, by looking how much red there is.

The point we are making is that the formation region has shifted from the periphery of the Greenland Sea Gyre during the earlier period, to the center of the gyre in the later period. To further clarify this, we state in the revision that the layer thickness of small σ_0 - π_0 distances outside the

Greenland Sea gyre (166 ± 172 m) is higher than that inside the Greenland Sea gyre (131 ± 154 m) in the earlier period, which is depicted graphically in Fig. 5a.

Reviewer #2 (Remarks to the Author):

This manuscript by Jie Huang et al. presents an analysis on the source and pathways of the densest component of the overflow waters from the Nordic Seas using a recently-developed metric of potential density - potential spicity distances. The manuscript is very well written, clearly structured with appropriate figures and the results are novel. I only have a few comments and therefore would recommend minor revisions before publication in Nature Communications.

Major comments

This study provides a very clear and thorough analysis of the sources and pathways of the densest component of the overflows. While reading, there were two aspects that I feel are missing in the discussion of the results.

First, a discussion is missing regarding pathway and source variability related to previous findings of e.g. Köhl 2010 (<https://www.tandfonline.com/doi/abs/10.1111/j.1600-0870.2009.00454.x>), de Jong et al. 2018 (<https://doi.org/10.1016/j.dsr.2018.07.008>) and Mauritzen 1996 ([https://doi.org/10.1016/0967-0637\(96\)00038-6](https://doi.org/10.1016/0967-0637(96)00038-6)). Previously, the Greenland Sea gyre was assumed to play a minor role for the overflows as the variability of open ocean convection was very large compared to the transport variability of the overflows. In this study, the focus is specifically on the Arctic-origin part of the overflow, considered to be mostly carried by the NIJ. How does the variability of this watermass at Denmark Strait and FBC compare to the dense water production in the Greenland Basin? Previous studies suggest that the strength of the NIJ is very variable and that export of the Greenland Sea watermass along the EGC is also a possibility (Köhl 2010, Messias et al. (<https://doi.org/10.1016/j.pocean.2007.06.005>)). How does this compare to your findings, as the EGC pathway is not found? Does the EGC pathway only exist for less dense waters of the Greenland Basin? Or is there still a possibility for the densest waters to leave the

Greenland Basin along the EGC, but are they substantially modified along the way and therefore no longer contribute the densest part of the overflow? Some elaboration on these aspects would strengthen the manuscript.

Thanks for this comment. With regard to the EGC pathway, we now state the following in the revised manuscript: “In Fig. 2d one sees that the shallowest depth of small σ_0 - π_0 distances along the continental slope of East Greenland is deeper than the sill depth of Denmark Strait (650 m). This implies that the Arctic-origin water located below the Atlantic-origin water in the EGC is too deep to contribute to the densest Denmark Strait overflow, in contrast to previous assertions (*Köhl* 2010 and *Messias et al.*, 2008). However, there is evidence of aspiration of Arctic-origin water just upstream of the sill (*Harden et al.*, 2016), which may reconcile these seemingly contrasting views.”

With regard to the variability of the densest Arctic-origin overflow water, we now discuss the seasonality of this water mass and include a new supplementary figure (Fig. S2). In particular, we constructed timeseries of the layer thickness of small σ_0 - π_0 distance water ($< 0.05 \text{ kg m}^{-3}$) in the upper 650 m, i.e. above the sill depth of Denmark Strait (results are indistinguishable using the Faroe Bank Channel sill depth of 850m). Four regions were chosen corresponding to the progression of the water emanating from the Greenland Sea gyre, as described earlier in the paper. In the gyre one clearly sees the wintertime formation of the water, while along the Mohn Ridge the newly ventilated water appears in spring/early-summer. However, there is no evidence of a seasonal signal along the two pathways bounding the Iceland Sea. We invoke hydraulics as a possible explanation, and include references. We note that as the dense water crosses the West Jan Mayen Ridge into the Iceland Sea, it presumably does so through gaps in the ridge. Waves carrying information about upstream time dependence will not get completely transmitted across the ridge if the gaps are hydraulically controlled. A detailed investigation of this, however, is beyond the scope of our study.

Second, a discussion is missing on how the findings compare to findings from Lagrangian observation and modelling studies regarding the origin of the overflows (e.g. Behrens et al. 2017 (<https://doi.org/10.1002/2016JC012158>)). Ln. 168 is a bit too limited in that respect.

We now state that previous studies give differing views regarding the origin of the NIJ. In particular, model simulations have suggested both the central Iceland Sea (*Våge et al., 2011*) and the southern Iceland Sea (*Behrens et al., 2017*) as source regions, while Lagrangian measurements imply an eastern source (*de Jong et al., 2018*). A recent study has revealed a bottom intensified current of dense overflow water progressing eastward along the north side of the Iceland-Faroe Ridge, presumably supplying the Faroe Bank Channel overflow (*Semper et al., 2020*). It remains unclear how our findings tie into these different studies, but we point out in the paper that it's likely that the two branches of Arctic-origin overflow water identified in our study feed both the NIJ and this newly identified current.

Minor comments

Ln. 22 and throughout the manuscript: For clarity I would rephrase 'how close two water parcels are to each other' to 'how similar two water parcels are in terms of physical properties'.

Done.

Ln. 24 and Ln 29: At reading only the abstract, it is not directly clear what is meant with the Greenland Sea gyre versus the periphery of the Greenland Sea gyre. Maybe replace 'gyre' with 'basin' or 'interior'?

Good point. We replaced “Greenland Sea gyre” with “the interior of the Greenland Sea gyre”.

Ln. 56: The ending of the introduction is a bit abrupt. Regarding the last sentence it seems as if the main aim of the paper is to investigate whether the densest component of the FBC overflow has a similar origin as the NIJ water mass. I would advice to make more clear what the main aim is of the paper and how the paper is structured.

We have added two sentences at the end of the introduction to clarify the aim of the paper.

Ln. 84: Can you explain what exactly you mean with 'a minor source' and how you conclude that from Fig. 2a? Number of measurements or magnitude of potential density - potential spicity distance does not necessarily imply how much source A contributes to the overflow and how much source B contributes in my understanding.

This is a good point, which was also raised by reviewer #1. In the revised manuscript we now estimate the rate of production of small σ_0 - π_0 distance water in the Greenland Sea Gyre. In particular, we constructed composite vertical sections across the center of the gyre for the autumn time period (before convection) and late-winter time period (after convection). Assuming a gyre radius of 150 km, the change in volume of the small σ_0 - π_0 distance water ($< 0.05 \text{ kg m}^{-3}$) between the two periods is $1.8 \pm 0.4 \times 10^4 \text{ km}^3$. This translates to a yearly formation rate of $0.6 \pm 0.1 \text{ Sv}$. We note that this estimated production is biased low because of the unaccounted export of dense water during the wintertime. However, it is comparable with the annual transport of the densest overflow in the NIJ as well that flowing across the sill at Denmark Strait (0.5 - 0.6 Sv , *Våge et al.*, 2011 and *Harden et al.*, 2016).

We also point out in the revised manuscript that only a small number of hydrographic profiles (5%) recorded small σ_0 - π_0 distances ($< 0.05 \text{ kg/m}^3$) in the late-winter mixed layers in the Iceland Sea.

Furthermore, most of the overflow water ventilated in the Iceland Sea is less dense than that formed in the Greenland Sea, and the winter mixed layers in the Iceland Sea only extend to a few hundred meters (as demonstrated in Supplementary Fig. 1). These results together imply that the Iceland Sea plays a relatively minor role in the production of the densest overflow water feeding the North Atlantic. This is all included in the revised manuscript.

Ln. 124: I am not sure whether 'evidence' is the right term here. The Eulerian velocity fields indeed show that the pathways suggested by the potential density - potential spicity distance analysis are likely, but a combination with a Lagrangian approach is needed to really prove the connectivity.

Agreed. The word “evidence” has been replaced by “support”.

Ln. 127: to resolve → to investigate? With 5 transects you're not resolving the circulation.

Agreed. The word “resolve” has been replaced by “elucidate”.

Ln. 174: Is the spatial and temporal distribution of the observations used for analysis in the early period comparable to the later period?

Yes. This is now stated in the revision.

Ln. 176: Are these variations small compared to the maximum potential density - potential spicity distances used to find the source regions? Please specify.

The change of the potential density and potential spicity of the NIJ water from 1990 to the present is 0.002 and 0.01 kg/m³, respectively. This change is small compared with the threshold of the small σ_0 - π_0 distance (0.05 kg/m³). This is now stated in the revision.

Ln. 213: to better understand

This change has been made.

Figures in general: Could you increase the font size of text and colorbar labels etc.? At the moment it is barely readable on a A4 printout.

All figures have been updated with increased font size of text and colorbar labels.

Fig. 2: It is difficult to distinguish the gray isobaths from the gray measurement circles, maybe use a different color?

The gray isobaths and the gray measurement circles are now distinguished more clearly in Fig. 2.

Fig. 3a-b: The red text is very difficult to read.

The red text is now easier to read in Fig. 3a-b.

REVIEWERS' COMMENTS:

Reviewer #1 (Remarks to the Author):

Thanks very much for time taken to respond to my questions and comments. The provided explanations and responses to my questions and comments are convincing and satisfying. From my personal opinion the manuscript has improved and reads very well, the story is clearly presented and of interested to a wider research community. I do not have any further questions, or comments.

Reviewer #2 (Remarks to the Author):

The authors have carefully addressed my comments. In particular, the addition of the seasonal evaluation of the layer thickness of small sigma-pi distances is really insightful. I recommend the paper is accepted for publication by Nature communications now.

Both reviewers are satisfied with the revised manuscript. There are no more questions raised by the reviewers. We appreciate the reviewers' previous comments and suggestions on the manuscript.

Reviewer #1 (Remarks to the Author):

Thanks very much for time taken to respond to my questions and comments. The provided explanations and responses to my questions and comments are convincing and satisfying. From my personal opinion the manuscript has improved and reads very well, the story is clearly presented and of interested to a wider research community. I do not have any further questions, or comments.

Reviewer #2 (Remarks to the Author):

The authors have carefully addressed my comments. In particular, the addition of the seasonal evaluation of the layer thickness of small sigma-pi distances is really insightful. I recommend the paper is accepted for publication by Nature communications now.